# SERS Detection of the Anti-Epileptic Drug Perampanel in Human Saliva

**DOI:** 10.3390/molecules28114309

**Published:** 2023-05-24

**Authors:** Matteo Tommasini, Andrea Lucotti, Luca Stefani, Sebastiano Trusso, Paolo M. Ossi

**Affiliations:** 1Dipartimento Chimica, Materiali, Ing. Chimica, Politecnico di Milano, 20133 Milano, Italy; matteo.tommasini@polimi.it (M.T.); andrea.lucotti@polimi.it (A.L.); 2Dipartimento Energia, Politecnico di Milano, 20133 Milano, Italy; luca1.stefani@mail.polimi.it; 3Istituto per i Processi Chimico Fisici, Consiglio Nazionale delle Ricerche, 98158 Messina, Italy; trusso@ipcf.cnr.it; 4Dipartimento Scienze Chimiche, Biologiche, Farmaceutiche ed Ambientali, Università degli Studi di Messina, 98166 Messina, Italy

**Keywords:** Surface Enhanced Raman Scattering (SERS), Therapeutic Drug Monitoring (TDM), Anti-Epileptic Drug (AED), Perampanel, human saliva

## Abstract

Surface-Enhanced Raman Scattering (SERS) can obtain the spectroscopic response of specific analytes. In controlled conditions, it is a powerful quantitative technique. However, often the sample and its SERS spectrum are complex. Pharmaceutical compounds in human biofluids with strong interfering signals from proteins and other biomolecules are a typical example. Among the techniques for drug dosage, SERS was reported to detect low drug concentrations, with analytical capability comparable to that of the assessed High-Performance Liquid Chromatography. Here, for the first time, we report the use of SERS for Therapeutic Drug Monitoring of the Anti-Epileptic Drug Perampanel (PER) in human saliva. We used inert substrates decorated with gold NPs deposited via Pulsed Laser Deposition as SERS sensors. We show that it is possible to detect PER in saliva via SERS after an optimized treatment of the saliva sample. Using a phase separation process, it is possible to extract all the diluted PER in saliva from the saliva phase to a chloroform phase. This allows us to detect PER in the saliva at initial concentrations of the order of 10^−7^ M, thus approaching those of clinical interest.

## 1. Introduction

Epilepsy is among the most diffuse neurological disorders, affecting between 0.6% and 1% of the world’s population [1]. An estimated 2.4 million new cases are diagnosed yearly, mostly in low- and middle-income countries. Epilepsy is a chronic disorder of the brain that manifests as recurrent seizures. These are episodes of partial to complete loss of consciousness, sometimes associated with loss of control of some functions of the body and muscle spasms, even involving the entire body. The frequency of seizures varies widely, from a few times a year to several per day. Perampanel (PER) is a recently developed Anti-Epileptic Drug (AED). PER was approved in 2012, and its efficacy and safety have been investigated since then. AEDs belong to the family of drugs with a Narrow Therapeutic Index (NTI). Indeed, the dose range in which the drug is effective without causing side effects is narrow. Patients who have received a dose that exceeds the upper limit of this range will experience side effects, while for patients who have received a dose below the lower limit, the drug is ineffective in controlling seizures. To avoid these problems, it is necessary to provide a tailored treatment that requires careful fine-tuning of the drug dosage for the patient, since the TI varies from person to person [2]. Therapeutic Drug Monitoring (TDM) is used for this purpose. TDM involves constantly measuring the concentration of the drug in the patient’s biological fluids, such as blood plasma [3]. High-performance liquid chromatography (HPLC) coupled with mass spectrometry (MS) is the most widely used laboratory technique for TDM. The measurements are accurate and reliable but are time-consuming and expensive. A standard test can take 1 to 4 days, and a single analysis costs between EUR 15 and EUR 30 [4]. This limits the population of patients who can access HPLC-MS and the frequency with which tests can be performed on a patient. Therefore, research focuses on developing techniques complementary to HPLC-MS to speed up TDM. Surface Enhanced Raman Spectroscopy (SERS) is among the most promising of these. Thanks to plasmonic mechanisms [5], SERS is a successful strategy to overcome the inherently weak signal of Raman spectroscopy, thus fostering a broad range of applications [6,7,8,9,10,11,12,13,14], including molecular sensing for TDM [15,16,17]. SERS was reported to detect low drug concentrations, demonstrating, under resonance conditions, an analytical capability comparable to HPLC [18]. SERS may be used in different fields of drug analysis, bringing significant advantages in sample preparation and measurement speed, with application to the analysis of active pharmaceutical ingredients and quality control of the production process [19]. By exploiting nanostructured metal surfaces irradiated with laser excitation of suitable photon energy, it is possible to increase the analyte Raman signal. If the analyte molecule has been adsorbed on the surface, the plasmonic enhancement will increase the intensity of its Raman signatures, thus allowing their detection.

In this work, we aim to demonstrate that it is possible to quantitatively detect the PER present in saliva using SERS sensors made of arrays of Au nanoparticles produced by Pulsed Laser Deposition (PLD), according to the procedure introduced in [20]. Saliva was considered for TDM of AEDs in the 1970s [21]. It is gaining interest, and it is increasingly used, given the potential advantages it has over serum or blood, mainly concerning sample collection. The sampling of saliva is much easier, since a specialized operator is not required, and the stress on the patient is minimized [22]. Moreover, the concentration of drugs in saliva generally reflects the pharmacologically active amount in the blood that is not bound to serum proteins. However, the use of saliva as a fluid for TDM has some disadvantages: the results obtained can be affected by contaminants present in the saliva, such as residues of other drugs or food, the volume of saliva that can be collected may be insufficient to perform clinical tests, and the viscosity of saliva can cause problems during pipetting [22]. Moreover, the concentration of the drugs in saliva can be lower than in blood plasma [23]. Indeed, the reported concentrations of PER in saliva are 2.3 × 10^−8^ M (total) and 8 × 10^−9^ M (free) [24], whereas the reported PER concentration in blood plasma at steady-state conditions is above ~2 × 10^−7^ M [25], and the reference concentration range in plasma is 2.5 × 10^−7^ M–2.8 × 10^−6^ M [26]. These figures make the SERS measurement of PER in saliva challenging. Given the exploratory nature of this study, we worked at drug concentrations such that the SERS signal of PER could be easily recognized from the background, without focusing on the limit of detection and the range of concentrations of clinical interest. We explored the most suitable conditions to detect PER in saliva samples containing controlled amounts of the drug. We addressed both the saliva sample treatment before the SERS analysis and the quantitative response of the sensors to varying concentrations of PER.

## 2. Results and Discussion

The conceptual scheme of the experiments described below is the following: First (a), we determined the Raman spectra of saliva to assess their relevant features. (b) Given that SERS of PER is fostered by protonation of the drug in an acidic environment [27], we checked the buffering ability of saliva, to properly overcome it by using suitable amounts of an acidic solution containing chlorides that we had previously tested [27]. (c) Since saliva already contains chlorides, we used a chloride-free acidic solution, which resulted in better SERS performance. (d) Hence, we could test the concentration-dependence of the SERS signal of PER in the range 1–0.1 mM, for which the PLD SERS sensors provide signals of acceptable intensity. The above range is rather far from the concentration range of clinical interest of PER in saliva, yet this series of experiments was necessary to check if the response of the sensors to changes in drug concentration was negatively altered by interfering species that could co-adsorb on the PLD sensors together with PER. Despite the sizeable data scatter, we could determine concentration-dependent SERS signals from PER, which prompted us to modify the saliva sample treatments by adopting a solvent extraction approach. (e) We chose chloroform, and by UV–Vis spectroscopy we determined that almost all the PER that was spiked into the saliva samples could be transferred to the chloroform phase. This result introduced the final step of our investigation, which aimed at proving the feasibility of detecting PER in saliva in a range of concentrations closer to those of clinical interest. The extraction with chloroform, followed by drying of the sample and successive re-dispersion in methanol under acidic conditions was successful in providing reliable SERS signals of PER down to the initial drug concentration in saliva of 10^−7^ M.

**(a) Spectroscopic properties of saliva.** Before attempting the detection of PER in saliva, we examined the Raman spectra of saliva samples dried on an aluminum foil. The samples were taken at different times on five different days, to assess the possible background signals of saliva and their variability. The results reported in Figure 1 show a limited variability of the main Raman signals of saliva, which are also consistent with data from the literature [28,29]. Moreover, apart from the Raman line at 1002 cm^−1^, none of the other peaks in the Raman spectra of saliva overlap with the expected PER markers we considered in the past [27].

**(b) Buffering properties of saliva**. We performed SERS experiments on saliva spiked with known amounts of PER, relying on the fact that the signals from saliva would not significantly interfere with the relevant signals of the drug. However, since the control of the (acidic) pH of the PER solutions subjected to SERS analysis was highlighted [27], we first had to characterize the pH of saliva under the addition of known quantities of the acidic solution that was employed to foster by protonation the SERS signal from PER. Three separate measurements of the pH of saliva were made using litmus paper (the samples were taken at different times on different days). The pH ranged between 6.25 and 7, in agreement with the literature [30]. In unstimulated saliva, three separate buffering systems provide it with a total buffering capacity of approximately 5.93 mmol H+/l [30]. The main buffer is based on bicarbonate chemistry and contributes to about 80% of the buffering capacity; the other buffers are based on proteins contained in saliva and phosphates. To determine the actual buffering capacity of saliva with the acidic solution used to protonate PER we added stepwise controlled volumes of acidic solution A to 1 mL of saliva. After each addition, the pH of the solution was measured with litmus paper. Table 1 shows that at least a 1:8 volumetric ratio (saliva/solution A) is required to bring the solution to pH 2, which was previously considered the optimal value for SERS measurements of PER [31].

Therefore, we spiked the solution obtained by mixing 10 μL of saliva with 80 μL of acidic solution A with 10 μL of a 10^−3^ M solution of PER in methanol. The PER concentration in the resulting solution was 10^−4^ M. In Figure 2, we display the SERS spectrum of such a solution, where the main signals of PER at 671, 880, 1000, and 2224 cm^−1^ can be clearly identified. We could not reliably recognize the PER markers at lower concentrations from the background.

**(c) Use of chloride-free acidic solution**. Moving from the consideration that the natural amount of chloride anions in saliva (3·10^−2^ M [32]) is enough to promote SERS, whereas an excess of chlorides may be detrimental to SERS [33], we modified the composition of the acid solution A used for the sample preparation. The resulting acid solution B is chloride-free and has a lower pH: this allows to control the pH of saliva with lower amounts of added acid solution. Before performing SERS, we tested again the pH conditions of the 10^−4^ M solution of PER obtained by mixing 180 μL of saliva with 20 μL of 10^−3^ M PER in methanol. By adding to such solution 2, 3, 4, and 5 μL of the acidic solution B we could control the pH at the values of 5, 4, 3 and 2, respectively. The pH of the solutions was measured with litmus paper. The SERS spectra of the four PER solutions above 10^−4^ M in saliva are reported in Figure 3 The main PER markers can be observed (compare, e.g., with Figure 2); however, the background fluctuates, being intense in the vicinity of 1600 cm^−1^ for pH 2 and 3. On these grounds, we decided for the following experiments to keep pH 4. At this pH value, we are still confident that PER is protonated [31], while at the same time the background is less disturbing than at lower pH values.

**(d) Concentration-dependent SERS of PER in saliva**. Based on the above findings, we attempted to quantify PER in saliva with a series of four concentration-dependent experiments at 10^−3^, 5 × 10^−4^, 2.5 × 10^−4^, and 10^−4^ M. The acidified saliva samples spiked with variable amounts of PER were prepared with the same procedure described above, and the composition is reported in Table 2. 

The SERS measurements on the samples described in Table 2 were performed using the procedure described in the Methods section. For each sample, we made five measurements at different spots on the SERS substrate to obtain the average value and the standard deviation of the SERS signal of PER. Representative SERS spectra for each PER concentration are displayed in Figure 4. The SERS intensities of the main markers of PER (671, 880, 1000, 2224 cm^−1^) were determined for each concentration after baseline correction and are listed in Table 3, together with the associated standard deviations. For each concentration, the data in Table 3 show a sizeable data scatter of the SERS response as a function of the measurement point. Such variations can be attributed to the rather intense background originating from compounds in saliva that are deposited on the substrate together with the analyte. The non-negligible effects of the background are also evident from the fact that the linear relationship between the SERS signals and the PER concentration does not tend to zero at null PER concentration due to the practical impossibility of eliminating the background signal (Figure 4b). A second issue is the relatively low sensor sensitivity to PER, which makes it difficult to determine drug concentrations below 10^−5^ M, whereas the expected concentration of PER in the patient’s saliva is several 10^−8^ M, which is still far from the concentration values we attained.

**(e) Detection of PER by solvent extraction from saliva.** Phase separation with water-immiscible chloroform is a convenient way to transfer PER from saliva to chloroform. Indeed, PER is significantly more soluble in chloroform than in water [34,35]. Hence, upon dispersing chloroform in water, we may expect a good transfer of PER from the water environment of saliva to chloroform. To test this approach, we prepared a solution of PER dissolved in saliva to which we added chloroform in a 1:1 volume ratio. The solution was magnetically stirred for 1 h. After centrifugation at 5000 rpm for 20 min (Figure 5a) we can distinguish the aqueous phase (top), a semi-solid phase consisting of the high-molecular-weight components of saliva (middle) and the chloroform phase (bottom).

The chloroform phase was extracted and diluted with pure chloroform in a volumetric ratio of 1:3 to obtain a clear solution, thus avoiding light-scattering phenomena that affect UV–Vis spectroscopy. The resulting spectrum in Figure 5b, by comparison with the reference spectrum of a PER chloroform solution, demonstrates the outcome of the extraction procedure, as witnessed by the peak at 350 nm, which is assigned to PER and falls in a region free from absorptions of other molecular species in the chloroform extract (compare with each other the red spectrum and the green spectrum in Figure 5b).

Hence, we assessed by UV–Vis spectroscopy the amount of PER that is transferred from saliva to chloroform. This was done by comparing two series of UV–Vis experiments on samples prepared according to the quantities reported in Table 4.

In the experiments of the first series (a) we prepared saliva samples spiked with known quantities of PER to extract PER with chloroform and to measure the UV–Vis spectrum of the extract. In the second series (b) we spiked with known amounts of PER the chloroform extract of as-is saliva to provide the reference absorbance of a known quantity of PER in the same chemical environment as that of series (a). The comparable absorbances reported in Figure 6 for the two series of experiments show that the extraction of PER from saliva to chloroform is nearly quantitative.

The possibility to effectively extract PER from saliva paves the way to detect PER with SERS on the chloroform extracts, which could improve the sensitivity of the technique since some of the chemical species contained in saliva, not being soluble in chloroform, would be separated before the SERS analysis and would not contribute to the background which has shown to be detrimental to the analysis (Figure 3). However, straightforward SERS detection of PER in the chloroform extract is not viable, because in this environment we cannot control the pH and the protonation of the drug, a relevant step to promote SERS. To solve this issue, we considered two additional steps after the phase separation process: (i) the chloroform extract is evaporated, and the solid residue is solubilized in methanol; (ii) the acidic solution A is added to the methanol solution to foster PER protonation. This approach was tested on two saliva samples spiked with PER at two different concentrations, namely (i) 5 × 10^–4^ M and (ii) 10^–7^ M. For both saliva samples, the extraction with chloroform was carried out as above described. For the saliva sample (i), 300 μL of chloroform extract were let dry, whereas for case (ii), we evaporated 1 mL of chloroform extract. The solid residues were redissolved in 100 μL (i) or 50 μL (ii) of methanol, to which we added 200 μL (i) or 50 μL (ii) of acidic solution A. Finally, a drop from each acidified solution was deposited on a SERS substrate, and the resulting spectra are reported in Figure 7. We notice that the different volumes considered for the extracts of sample (i) and (ii), and for the successive steps, reflect the different initial concentrations of PER in saliva.

For both extracts, all the relevant PER markers are clearly visible in the SERS spectra. We therefore conclude that, by solvent extraction from saliva samples, we can overcome the sensitivity limitation which was evident in the first experiments we discussed (see Figure 2 and Figure 3).

## 3. Methods

The PLD SERS sensors were deposited on Corning glass supports with an optimized procedure aimed at obtaining sensors with repeatable lateral uniformity of their optical properties within any given sensor, among sensors from the same batch, and among sensors from different batches, as described in [36]. The typical morphology of the surface of a representative sensor is shown in Figure 8. The surface plasmon resonance of the SERS sensors used in this work is between 670 nm (1.86 eV) and 720 nm (1.72 eV).

The SEM image of Figure 8 was acquired in the NanoLab (Energy Dept., Politecnico di Milano) by a Zeiss Supra 40 Field-Emission Scanning Electron Microscope (FE-SEM) (Oberkochen, Germany), operating in high-vacuum and equipped with the GEMINI column.

The saliva samples were collected from a consenting healthy male volunteer (L.S.) under unstimulated conditions by spitting into a test tube. As discussed in the Results and Discussion section, some of the sample preparations required a centrifuge. We used an EBA 21 Hettich centrifuge (equipped with a swing-out rotor 1115, maximum capacity 6 × 35 mL) at the maximum angular speed of 5000 rpm.

The acidic solution A was prepared by adding 0.251 mL of sulfuric acid 96% (CAS:7664-93-9, Carlo Erba Reagents Srl, Cornaredo, Milan, Italy) and 41 μL of hydrochloric acid 37% (CAS:7647-01-0, Sigma-Aldrich, Milan, Italy) to 500 mL of distilled water. The measured pH of solution A (litmus paper) was 2. The acidic solution B was prepared by adding 2.783 mL of sulfuric acid 96% (CAS:7664-93-9, Carlo Erba) to 50 mL of distilled water. The measured pH of solution B was 0.

All the SERS spectra were measured using the dispersive Raman spectrometer Horiba Jobin Yvon LabRAM HR8000 (Kyoto, Japan), equipped with a 600 grooves mm^−1^ grating, an Olympus BX41 microscope, and a Peltier-cooled CCD detector. For the spectroscopic measurements, an analyte droplet of approximately 1–2 μL was deposited on the SERS sensor and air-dried. A 633 nm He:Ne laser, focused through a 50× objective, was used as the excitation source. The laser power was 0.3 mW; the exposure time and the number of averaged exposures could vary depending on the experiment and are reported on a case-by-case basis.

The UV–Vis spectra reported in this work were measured with the JASCO V-570 spectrophotometer (Jasco, Tokyo, Japan) using quartz cuvettes with an optical path of 1 cm.

## 4. Conclusions

In conclusion, we proved that it is possible to detect by SERS the antiepileptic drug Perampanel solubilized in human saliva. We used sensors made of arrays of gold nanoparticles produced by pulsed laser deposition. The SERS signal of Perampanel is directly proportional to the drug concentration in the salivary solution in the concentration range 10^−3^–10^−4^ M. A phase separation process based on chloroform and methanol allowed us to detect Perampanel down to the initial concentration in saliva of 10^−7^ M, which approaches the range of concentration of clinical interest for this application of SERS. Considering the volume ratio between the chloroform extract and the methanol used to redissolve the solid residue from chloroform, it will be possible to optimize the extraction procedure. One remarkable aspect of the above-proposed strategy is the possibility of obtaining a solution of PER diluted in methanol that is more concentrated than the starting solution in chloroform. As an example, by drying 1 mL of a chloroform solution of PER and redissolving the solid residue with 10 μL of methanol, it is possible to obtain a solution with a PER concentration increased by a factor of 10^2^.

## Figures and Tables

**Figure 1 molecules-28-04309-f001:**
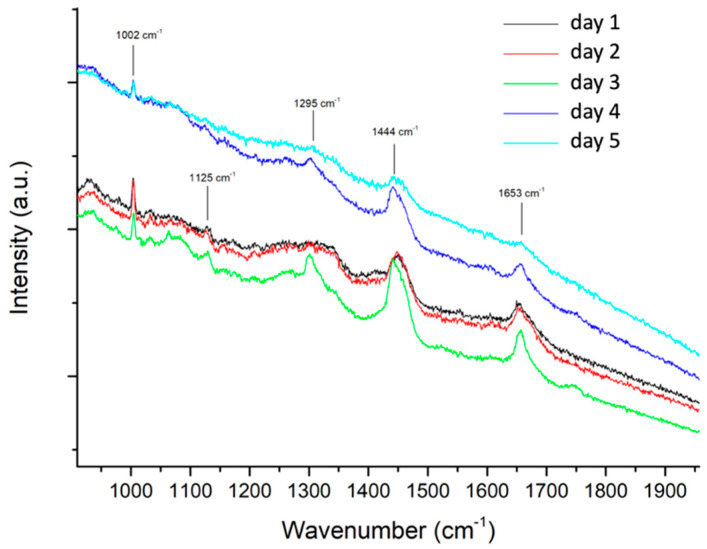
Raman spectra of dried saliva samples taken at different times on five different days. Each spectrum was collected on an aluminum foil using 785 nm excitation, 1 mW power, 50× objective (average of 3 collections, each of 30 s).

**Figure 2 molecules-28-04309-f002:**
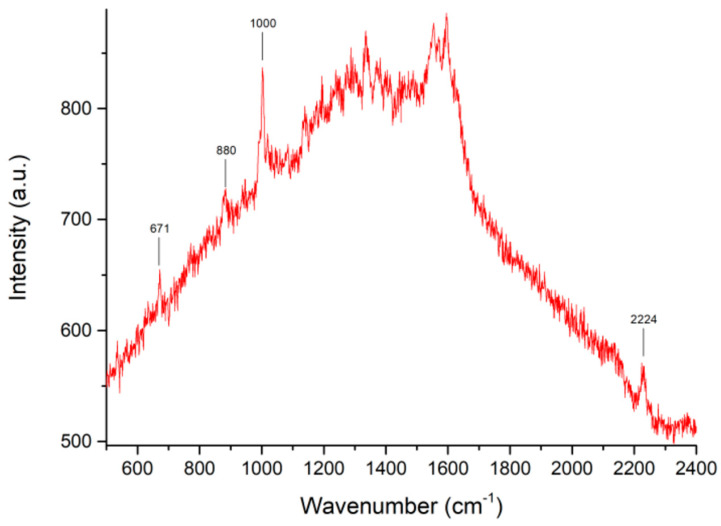
SERS measurement of PER diluted in saliva in controlled acidic conditions (pH 2, see text). The concentration of PER is 10^−4^ M. Instrumental parameters: laser wavelength 633 nm; laser power 0.3 mW; 50 × objective. The reported spectrum is the average of three exposures of 60 s each.

**Figure 3 molecules-28-04309-f003:**
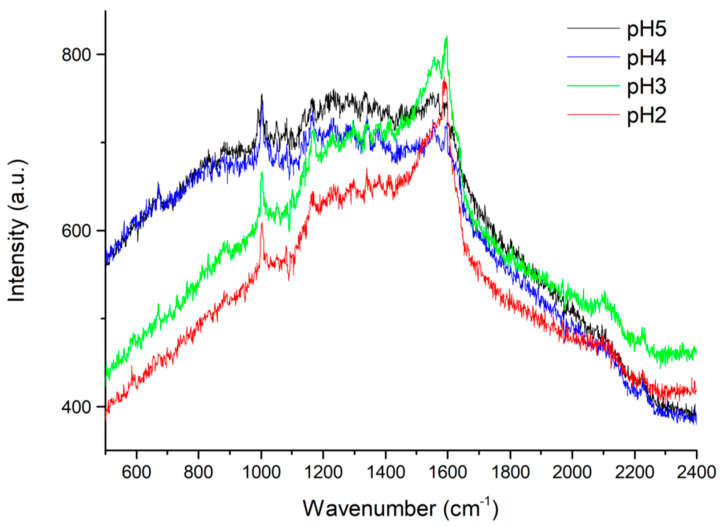
SERS of PER in saliva as a function of pH. PER concentration, 10^−4^ M. Instrumental parameters: laser wavelength 633 nm; laser power 0.3 mW; 50× objective. Each reported spectrum is the average of 10 exposures of 60 s each.

**Figure 4 molecules-28-04309-f004:**
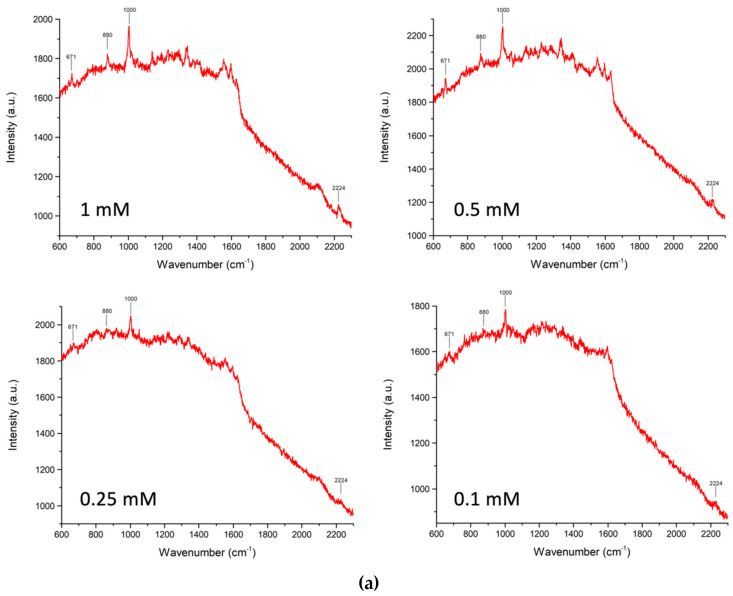
(**a**) Representative SERS spectra of acidified saliva samples spiked at known PER concentrations (see text for details). (**b**) Concentration-dependence of the average SERS intensities of the four relevant markers of PER (we carried out five independent measurements for each concentration—see Table 3; the dots indicate the average values of the five independent measurements and the error bars indicate Student’s 95% confidence intervals).

**Figure 5 molecules-28-04309-f005:**
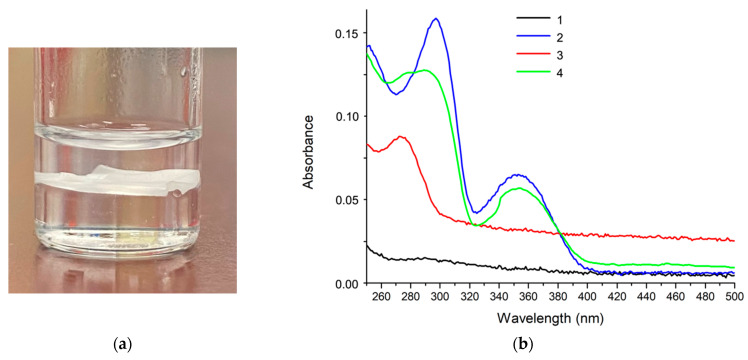
(**a**) Phase separation observed after the settling by centrifugation of a saliva sample previously shaken with chloroform in a 1:1 volume ratio. From the bottom to the top of the vial: the chloroform-rich phase, the semi-solid intermediate phase, and the water-rich phase. (**b**) UV–Vis spectra of chloroform after being contacted with pure water (1, black); PER in chloroform (2, blue); the chloroform extract of as-is saliva (with no added PER, 3, red); the chloroform extract of the saliva sample spiked with PER (green, 4).

**Figure 6 molecules-28-04309-f006:**
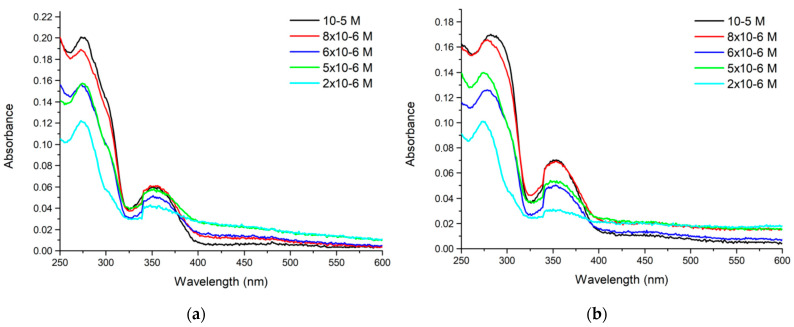
(**a**) UV–Vis spectra of the chloroform extracts of saliva samples spiked with PER at the reported concentrations. (**b**) UV–Vis spectra of the chloroform extracts of as-is saliva, later spiked with known quantities of PER.

**Figure 7 molecules-28-04309-f007:**
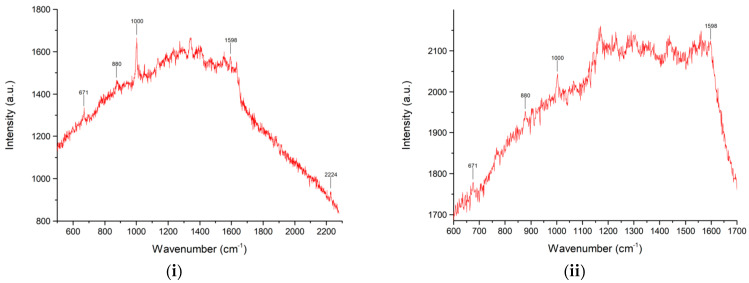
SERS spectra of solvent-extracted samples obtained from saliva spiked with PER at the concentration of (**i**) 5 × 10^–4^ M and (**ii**) 10^–7^ M. See text for details about the sample preparation for SERS analysis. Instrumental parameters: laser wavelength 633 nm; laser power 0.3 mW; 50× objective. The reported spectra are the average of three exposures of 120 s each.

**Figure 8 molecules-28-04309-f008:**
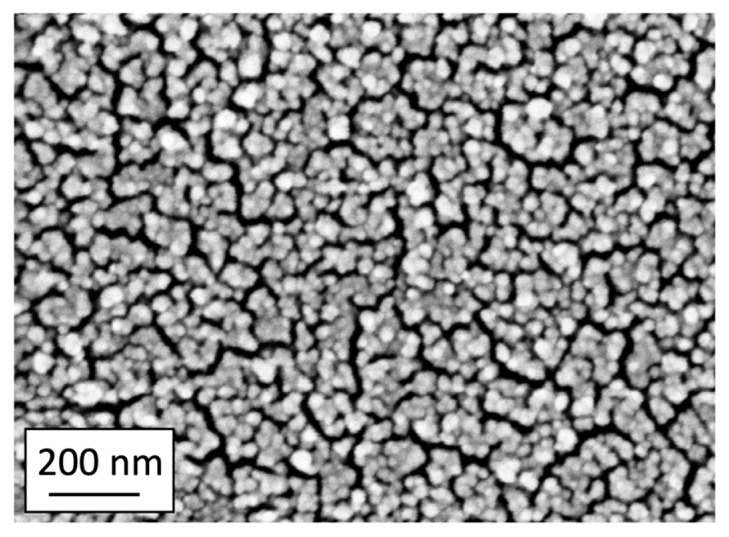
Representative SEM micrograph of a PLD SERS sensor. The lateral uniformity of the array of Au nanoparticles is evident.

**Table 1 molecules-28-04309-t001:** Changes in the saliva pH after addition with increasing quantities of acidic solution A.

**Sample Composition** **(Volume Ratios: Saliva/Solution A)**	**pH**	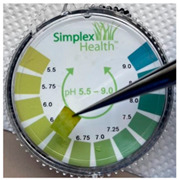 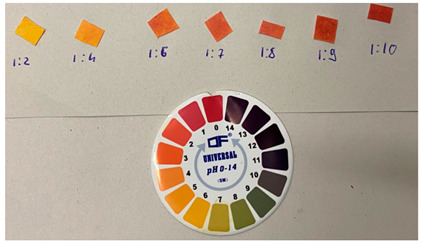
saliva	6–7	
1:2	5	
1:4	4	
1:6	3	
1:7	3	
1:8	2–3	
1:9	ca. 2	
1:10	2	
Acidic solution A	2	

**Table 2 molecules-28-04309-t002:** Composition of the saliva solutions (pH 4) containing PER in the 10^−3^–10^−4^ M concentration range. The pH was measured using a litmus paper. See text for details.

PER Solution in MeOH (μL, Either 10^−2^ or 10^−3^ M)	Saliva (μL)	Acidic Solution B (μL)	Final PER Concentration (M)	pH
10 (10^−2^ M)	90	0.6	10^−3^	4
100 (10^−3^ M)	100	1	5 × 10^−4^	4
50 (10^−3^ M)	150	1.1	2.5 × 10^−4^	4
20 (10^−3^ M)	180	1.2	10^−4^	4

**Table 3 molecules-28-04309-t003:** Concentration-dependent baseline-corrected SERS intensities (determined as areas under the peaks) of the relevant PER markers in the acidic saliva solutions (compositions in Table 2).

c (mM)	671 cm^−1^	880 cm^−1^	1000 cm^−1^	2224 cm^−1^	
1.0	2.6	5.9	21	7.8	
	6.5	7.2	20.3	3.1	
	7.7	13.4	22.4	9.2	
	6.1	9.1	22	9.1	
	4.3	8.1	21.5	5.2	
	5.4	8.7	21.4	6.9	average
	2	2.9	0.8	2.7	st. dev.
0.5	4.5	9.6	25	6.3	
	7.1	7.8	16.7	5.8	
	5.5	7.6	18.4	6.3	
	5.4	6.9	13.1	6.2	
	4.7	8.9	17.9	6.1	
	5.4	8.1	18.2	6.2	average
	1.1	1.1	4.3	0.2	st. dev.
0.25	3.5	6.4	11.4	5.4	
	5.2	7.3	12.7	4.8	
	6.6	5.6	13.6	4.7	
	3.7	4.1	12.7	4	
	3.6	5	12.7	3.2	
	4.5	5.7	12.6	4.4	average
	1.4	1.2	0.8	0.9	st. dev.
0.1	3.5	4.4	12.2	5.5	
	6.1	5.8	15.9	5.7	
	3.8	5.7	11.2	5.9	
	4.1	5.1	9.3	5.2	
	4.5	3.7	10	4.9	
	4.4	4.9	11.7	5.4	average
	1	0.9	2.6	0.4	st. dev.

**Table 4 molecules-28-04309-t004:** Compositions of (a) the saliva solutions spiked with controlled quantities of PER, and (b) the chloroform extracts of as-is saliva, later spiked with controlled quantities of PER. Procedure (a) 1. Prepare the saliva/PER solution; 2. Add chloroform in 1:1 volume ratio; 3. Stir for 1 h; 4. Centrifuge at 5000 RPM for 20 min; 5. Sample the chloroform phase; 6. Add chloroform in 1:3 volume ratio to the solution sampled in step 5.

10^−3^ M PER in MeOH (μL)	Saliva (μL)	PER Concentration at Step 1 (M)	PER Concentration at Step 6 (M)
(a)			
80	1920	4 × 10^−5^	10^−5^
64	1936	3.2 × 10^−5^	8 × 10^−6^
48	1942	2.4 × 10^−5^	6 × 10^−6^
40	1960	2 × 10^−5^	5 × 10^−6^
16	1984	8 × 10^−6^	2 × 10^−6^
(b)			
20	480	4 × 10^−5^	10^−5^
16	484	3.2 × 10^−5^	8 × 10^−6^
12	488	2.4 × 10^−5^	6 × 10^−6^
10	490	2 × 10^−5^	5 × 10^−6^
4	496	8 × 10^−6^	2 × 10^−6^

Procedure (b): 1. Sample saliva; 2. Add chloroform in 1:1 volume ratio; 3. Stir for 1 h; 4. Centrifuge at 5000 RPM for 20 min; 5. Sample the chloroform phase; 6. Add known volume of 10^−3^ M PER in MeOH to the solution sampled in step 5; 7. Add chloroform in 1:3 volume ratio to the solution.

## Data Availability

Not applicable.

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
