# Peer review of "SERS Detection of the Anti-Epileptic Drug Perampanel in Human Saliva"

_molecules, 2023, doi:10.3390/molecules28114309_

Round 1

Reviewer 1 Report

The purpose of this work is to develop an easy method to constantly detect the AntiEpileptic drug Perampanel in human saliva using SERS to replace the common method based on HPLC. After reading through the whole manuscript, I have to say that this work is not suitable for publication in its current form due to the following reasons.

(1)     The manuscript is not prepared according to a high standard. There are numerous writing and formatting issues that the authors must fix. For example, in table 1, the image of pH indicator is apparently stretched and distorted; in figure 5 caption, they use a and b to denote the sub-figures, yet they use i and ii to denote sub-figures in figure 8.

(2)     In reference section, the style used is not the common style used in Molecules.

(3)     The writing is misleading and did not properly reflect the limitations of this work. It seems that this work demonstrate the possibility to use chloroform-methanol extraction method to detect low-concentration anti-Epileptic drug Perampanel (PER) in saliva. Why the authors did not go one step further to detect this drug in real clinical samples after they think they found an optimized approach? The reproducibility of this work is not good. This is actually a long-standing limitation of SERS. When the laser probes different sample spots on SERS substrate, people get different SERS signal. The authors should mention this limitation in the manuscript. The standard deviation is quite large as shown in Table 3. The limitation of the reliability of this work must be emphasized in the abstract and the conclusion. In my mind, I think the current SERS method is not suitable for clinic at this point.

Language by itself is OK.

Reviewer 2 Report

In this manuscript, the authors reported a SERS sensor for detecting the anti-epileptic drug Perampanel (PER) in human saliva. However, there are some specific comments for this manuscript:

1. For Figure 2, the authors should indicate the according sample of each spectrum.

2. For Figure 5, the authors are suggested to show the error bar in the plots. Also, they should include 2-3 more concentrations for the detection and estimate the LOD.

3. In this manuscript, the authors claimed that the reported concentrations of PER in saliva are 2.3x10-8 M (total) and 71 8x10-9 M (free). However, the strategy proposed by the authors could only detect PER at 10−7 M in saliva. Could the authors figure out how to improve the detection performance to achieve 10-8 M?

4. There are some relevant papers about SERS sensors (e.g., Theranostics 12 (13), 5914; ACS Applied Materials & Interfaces 14 (3), 4714-4724). The authors are suggested to cite these papers in proper positions.

Minor editing of English language required

Reviewer 3 Report

The manuscript report for the first time the use of SERS for Therapeutic Drug Monitoring of the Anti-Epileptic Drug Perampanel (PER) in human saliva. And used inert substrates decorated with gold NPs deposited via Pulsed Laser Deposition as SERS sensors. Although this manuscript has achieved some new results, it requires significant revisions before accepted.

1.     The Raman spectrum acquisition of Fig.2 is completed under the condition of 785 nm laser, and there is almost no fluorescence background. Why the data behind are selected under the 633 laser acquisition, this time there is a very obvious fluorescence background peak, which greatly affects the real detection results. In my opinion, all the data should be collected under the 785 laser is more reasonable.

2.     In data analysis, the author cannot use only one result for statistics, and at least three parallel experiments are needed to exclude contingency, such as Figure 5.

3.     Some typical studies on SERS detection should be discussed (Angewandte Chemie International Edition, 58, 16523-1652, Advanced materials, 2018, 30, 1702275).

no

Round 2

Reviewer 1 Report

Please further improve the format quality of this manuscript during the editorial stage as I still think the quality of presentation is low.

OK

Reviewer 2 Report

The authors have answered my questions properly.

Reviewer 3 Report

The author has made appropriate modifications and can be published

No